# Biofunctionalization with a TGFβ-1 Inhibitor Peptide in the Osseointegration of Synthetic Bone Grafts: An In Vivo Study in Beagle Dogs

**DOI:** 10.3390/ma12193168

**Published:** 2019-09-27

**Authors:** Andrea Cirera, Maria Cristina Manzanares, Pablo Sevilla, Monica Ortiz-Hernandez, Pablo Galindo-Moreno, Javier Gil

**Affiliations:** 1Oral Surgery and Implant Dentistry Department, School of Dentistry, University of Granada, Campus Universitario La Cartuja s/n, 18071 Granada, Spain; pgalindo@ugr.es; 2Human Anatomy and Embryology Unit, DPyTEx, School of Medicine, University of Barcelona, 08036 Barcelona, Spain; mcmanzanares@ub.edu; 3Universitary School Salesiana de Sarrià—EUSS Autonomous University of Barcelona, 08017 Barcelona, Spain; Psevilla@euss.es; 4Biomaterials, Biomechanics and Tissue Engineering group (BBT), EEBE, Technical University of Catalonia (UPC), 08019 Barcelona, Spain; Monica.ortiz-hernandez@upc.edu; 5Bioengineering Institute of Technology, School of Dentistry, Universitat Internacional de Catalunya, 08195 Barcelona, Spain

**Keywords:** osseointegration, biofunctionalization, histomorphometrical analysis, scanning electron microscopy, TGF-β1, synthetic bone graft

## Abstract

Objectives: The aim of this research was to determine the osseointegration of two presentations of biphasic calcium phosphate (BCP) biomaterial—one untreated and another submitted to biofunctionalization with a TGF-β1 inhibitor peptide, P144, on dental alveolus. **Materials and Methods:** A synthetic bone graft was used, namely, (i) Maxresorb^®^ (Botiss Klockner) (*n* = 12), and (ii) Maxresorb^®^ (Botiss Klockner) biofunctionalized with P144 peptide (*n* = 12). Both bone grafts were implanted in the two hemimandibles of six beagle dogs in the same surgical time, immediately after tooth extraction. Two dogs were sacrificed 2, 4, and 8 weeks post implant insertion, respectively. The samples were submitted to histomorphometrical and histological analyses. For each sample, we quantified the new bone growth and the new bone formed around the biomaterial’s granules. After optical microscopic histological evaluation, selected samples were studied using backscattered scanning electron microscopy (BS-SEM). **Results:** The biofunctionalization of the biomaterial’s granules maintains a stable membranous bone formation throughout the experiment timeline, benefitting from the constant presence of vascular structures in the alveolar space, in a more active manner that in the control samples. Better results in the experimental groups were proven both by quantitative and qualitative analysis. **Conclusions:** Synthetic bone graft biofunctionalization results in slightly better quantitative parameters of the implant’s osseointegration. The qualitative histological and ultramicroscopic analysis shows that biofunctionalization may shorten the healing period of dental biomaterials.

## 1. Introduction

Grafting materials are currently a predictable treatment for regenerating bone defects in implant dentistry. There are several types of bone grafts, and autogenous bone grafting is the gold standard for its osteoconductive and osteoinductive properties [1,2,3]. Despite the properties of autogenous bone graft, however, not all cases present the desirable amount of bone available [4] and it is necessary to harvest the graft from a donor area, which may require a second surgical site [5]. Alternatively, synthetic biomaterials have emerged since the 1980s [6]. Reynolds et al. [7] performed a systematic study in which the researchers did not find differences in clinical response between granulate bone allograft and calcium phosphate ceramic biomaterial grafts.

Due to bone tissue remodeling, the hard tissue in alveolus presents a certain atrophy. The resorption of alveolar ridge dimensions can occur in 12 months [8]. Many biomaterial strategies have been developed to maintain the maximum amount of natural hard tissues, such as narrow dental implants. It is very important to preserve the bone volume after tooth extraction and supply a stable substrate for implant placement [9,10].

An important factor concerning socket preservation is angiogenesis. Angiogenesis is a complicated mechanism, including extracellular matrix (ECM) degradation, adhesion, proliferation, migration, and differentiation cells. [11]. The use of either autologous platelet-rich fibrin (PRF) or beta-tri-calcium phosphate with collagen (β-TCP-Cl) has been found to be excellent in socket preservation [12]. β-TCP has also been compared to β-TCP with added collagen, resulting in no statistically significant differences [13,14].

To further optimize the osteogenic properties of biphasic calcium phosphates (BCPs), the combination of BCPs with molecules or growth factors is being investigated [15]. Minier et al. [16] utilized local bone morphogenetic protein type 2 (BMP-2) release in BCP ceramic particles in a dog animal model. They observed that BMP-2 loaded grafts induced the formation of lamellar bone tissue with an excellent mineralization, well-differentiated and surrounding the BCP biomaterials and bridging granules/bone tissue interfaces eight weeks after surgical operation. Platelet-rich fibrin has also been used in combination with β-TCP, increasing the bone regeneration in the first two weeks after surgery [17].

BCP bone substitutes appear as an effective vehicle for osteoinductive molecules (bioactive peptides, bone growth factors) or even stem cells in a tridimensional structure, thus facilitating bone regeneration [18,19]. An adequate adjustment of both dosage and release profile of the bioactive molecule is required [20].

On the other hand, inflammatory and immune responses have been observed during the osseointegration process [21,22]. Transforming growth factor beta 1 (TGF-β1) has been signaled as the central mediator of the fibrotic response [23]. Moreover, it increased expression in the tissues around failed dental implants, among other roles [23,24,25,26].

A TGF-β1 inhibitor peptide, P144, blocks the interaction of TGF-β1 with its receptor, thus altering its interaction with different cell types [27,28,29]. An increase of osteoblast and osteoprogenitor cell differentiation was observed as the in vitro results of P144 biofunctionalization of a CP-Ti surface [30]. Previous in vitro studies in MC3T3-E1 stem cells showed improvement of the quality of interfacial interactions during the healing process [31]. Then, inhibiting TGF-β1 can either limit the formation of fibrous tissue around a biomaterial or activate the cells responsible for osseointegration [32,33].

The aim of this study was (i) to evaluate the effect on osseointegration of the biofunctionalization with a TGF-β1 inhibitor peptide (P144) of a well-known biomaterial and then (ii) to compare its osseointegration and histomorphometric values with the ones from the same untreated biomaterial. Osseointegration is considered as the union between bone and a biomaterial with mimetic properties. Osteoconduction according to Weber [34] is a three-dimensional (3D) process of ingrowth of sprouting capillaries, perivascular tissue, and osteoprogenitor cells from a bony bed into the 3D structure of a porous implant used as a guiding cue to bridge a defect with bony tissue.

## 2. Results

The postoperative healing was uneventful and peri-implant mucosa appeared to be clinically healthy. A total of 24 dental alveoli were processed, 22 of which were finally analyzed due to artifacts and process mishaps.

Bone index contact results were determined and can be seen in Table 1 for each time after implantation and each biomaterial.

Figure 1 shows the results of percentage of bone inside the alveolar defects during 2, 4, and 8 weeks after surgery.

The control biomaterial presents a difference in bone growth between weeks 2 and 4 (*p* = 0.003, marked as *), as well as 2 and 8 (*p* < 0.001, marked as *), and the difference between weeks 4 and 8 (*p* = 0.004, marked as *).

The P144-functionalized group shows higher values than the biomaterial alone throughout the experiment. Bone percentage presents statistical differences between 2 and 8 weeks (*p* < 0.001, marked as &) and between 4 and 8 weeks (*p* = 0.004, marked as $). However, no difference was observed between 2 and 4 weeks (*p* = 0.087).

The differences for the same time of implantation between the control and biofunctionalized biomaterial have been described with letters. The results of the bone percentage are higher for the biofunctionalized biomaterial. At 2 weeks, the differences present significance (*p* = 0.0007, marked as a) and at 8 weeks (*p* = 0.003, marked as b). However, no difference was observed at 4 weeks.

Figure 2 shows percentages of bone ingrowth around the biomaterial fragments.

The control biomaterial fragment values show differences between 2 and 4 weeks (*p* = 0.005, marked as *), between 4 and 8 weeks (corrected *p* = 0.029, marked as *), and between 2 and 8 weeks (*p* < 0.001, marked as *).

The P144-biofunctionalized group shows the highest values but no significant differences between 2 and 4 weeks in the growth between times. The significant differences are between 2 and 8 weeks (*p* < 0.001, marked as &) and between 4 and 8 weeks (*p* = 0.0025, marked as $).

The differences for the same time of implantation between the control and biofunctionalized biomaterial have been described with letters. The results of the bone percentage are higher for the biofunctionalized biomaterial. At 2 weeks, the differences present significance (*p* = 0.0008, marked as a) and at 8 weeks (*p* = 0.0041, marked as b). However, no difference was observed at 4 weeks.

### Histology and BS-SEM Analysis

The control group biomaterial is surrounded at a distance by sparse, thin, calcified trabeculae, mainly constituted by chondroid bone as can be observed in Figure 3a. With higher magnification, Figure 3c shows its characteristic big, irregular, confluent cell lacunae (orange dots) immersed in barely calcified extracellular matrix (deep green), lined by a band of non-calcified extracellular matrix (orange in Figure 3c) constituting thin trabeculae that establish sparse contacts between themselves and with the biomaterial granules. The thickness of the trabeculae in contact with the biofunctionalized biomaterial granules is greater than in the control group (Figure 3b,d), and its homogeneous deep green staining and separated polygonal cell lacunae show that they are constituted by more mature tissue and woven bone. The extracellular non-calcified matrix is visible around the bone cell lacunae from the newly formed tissue surrounding the trabeculae (Figure 3d).

The BS-SEM analysis (Figure 4a,c) reveals a similar bone ingrowth in both groups at two weeks after grafting at lower magnification. No apparent contact is visible between the alveolar cortical and the biomaterial fragments. However, when increasing magnification, it can be observed that in the peptide group (Figure 4d) most fragments show osseous tissue apposition, all situated near the preexisting cortical, whereas only some fragments of the control group present bone apposition (Figure 4b).

Four weeks after implantation, in the control samples, a dense mesh of thicker and denser trabeculae can be observed surrounding the biomaterial fragments and establishing contacts with the bone cortical (Figure 5a). The preexisting cortical area also shows evidence of an intense remodeling process, consisting in numerous, convoluted vascular channels and new lamellar bone apposition within (red arrows in Figure 5b). Higher magnification shows that new bone formation (orange, Figure 5d) is still evident lining the vascular channels (area in Figure 5c) that connect the new trabeculae with the cortical alveolar wall.

At this stage, the fragments of the biofunctionalized group adopt the same deep green color as the bone tissues of the trabeculae that surround them (Figure 6a,c). These denser and thicker trabeculae present numerous large interconnected vascular channels visible in the bone cortical (framed area in Figure 6b), in the alveolar space (framed area in Figure 6d), and in the periosteal reaction.

The higher magnifications reveal that the blood vessels are around the biomaterial granules while maintaining connections with the vascularization of the surrounding areas (Figure 7). The vessels (erythrocytes (e) are visible in Figure 7c) also invade the spaces within the biomaterial granules and initiate the bone tissue apposition process evidenced by the linear deposit of non-calcified extracellular osseous tissue matrix (orange in Figure 7b,c).

A different view at the same magnifications gives a more detailed image, with osteoblasts (red arrows in Figure 8b) already immersed in the still non-calcified extracellular matrix, within which the calcification starts to appear as sparse, still unconnected deep green granules. These images reveal the size of the vessels within the cavities of the biomaterial granules; limited amounts of erythrocytes (e) are visible in the biggest vessels (Figure 8a,b), whereas in the smaller cavities the erythrocytes appear piled in very small vessels (Figure 9b). In a similar manner, the tissues deposited show the morphological characteristics of the woven bone (Figure 9a)—big, isolated cell lacunae in a dense, very calcified extracellular matrix, and lamellar bone apposition proven by the osteoblasts (red arrows in Figure 9a) within the still undecalcified perivascular bone matrix (orange). Very different are the confluent and irregularly-shaped cell lacunae immersed in an extracellular matrix, whose calcification is still incomplete and slightly granular, which is the typical aspect of the chondroid bone shown in Figure 8b and Figure 9b.

In reference to the BS-SEM results, most of the P144-biofunctionalized granules show a solid and continuous contact with the cortical of the alveolar defect, whereas the control group has a more discontinuous connection, both between the fragments and the trabeculae and between the trabeculae and the preexisting cortical (Figure 10a,c). In the experimental group (Figure 10d), trabeculae are denser and show a more advanced phase of bone remodeling than the one observed in the non-biofunctionalized graft groups (Figure 10b). The bone apposition in the peptide group has a lamellar and continuous apposition upon the initial chondroid bone, whereas the control group still shows more chondroid bone and less lamellar bone.

Eight weeks after the biomaterials’ insertion, there is a general increase of the osseointegration in all study groups, coherent with the tendency showed in the results of the histomorphometric data.

Light microscopy analysis reveals that the bone architecture around all biomaterial fragments (Figure 11) is denser because of the apposition of lamellar bone lining the vascular channels. A higher number of vascular channels is visible between the biomaterial fragments in the control group (Figure 11b) than in the biofunctionalized material (Figure 11e). Moreover, the vessels connecting the preexisting cortical (pc in Figure 11b) are bigger and more convoluted in the control samples. When surrounding biofunctionalized fragments, the bone tissues are denser and more continuous with the preexisting cortical. The higher magnifications (Figure 11c,f) reveal that there are still numerous vascular channels around the trabeculae surrounding the biomaterial control fragments, whereas the trabeculae around the functionalized fragments are denser, with less and thinner vascular spaces. It is noteworthy, however, that the alveolar space in both groups still contains fragments of biomaterial that are not in contact with the recently deposited osseous tissues.

BS-SEM analysis shows a more mature stage of bone formation in both experimental groups. Figure 12a,c shows a dense and compact bone formation connecting the granules; in the treated group, this linking seems more consistent because the trabecular structure connects both corticals (framed areas in Figure 12a,c) of the alveolar defect and the grey color of the lamellar bone apposition is predominant. However, in the control group, the trabecular web is less continuous, and osteonal remodeling is visible mainly in the preexisting cortical (asterisks in Figure 12b). The peptide group shows osteonal apposition mainly within the biomaterial granules (Figure 12d).

## 3. Discussion

The purpose of this investigation was to compare the effects on bone ingrowth and osseointegration of a biofunctionalized synthetic bone graft with respect to a well-known synthetic bone graft. The bone grafts were inserted in the fresh extraction sites of the mandibular premolar area of beagle dogs the same day of the atraumatic teeth extraction. The aim of bone grafting is to limit ridge resorption and bone loss, as well as to preserve as much bone mass as possible for implant placement.

In vivo studies proved that calcium phosphates (CaPs) are biocompatible and offer an effective possibility to allograft biomaterial without the common problems such as an immune reaction and disease transmission [35,36]. The use of alloplastic materials can offer a useful procedure to avoid these risks [5]. The osteoconductive properties of BCP bioceramics have been enhanced with osteoinductive features via the modulation of its physicochemical properties [35]. The implanted calcium phosphate surface adsorbs the proteins that attract the osteoblasts, inducing the bone cells to differentiate and promoting the mineralization process [37].

Bone remodeling takes place around the BCP grains starting with the resorption of its calcium phosphate components. Thus, the integration of the granules of the biomaterial with the bone structure is continuously in progress. This interface is highly dynamic, involving the bone biology and physiopathology, the underlying biomechanical factors, and the niche factor (the surrounding vessels, biochemical factors, nutritional status, etc.). Bone remodeling is aimed at a “*restitutio ad integrum*”, resulting in an organized and mineralized bone ingrowth at the expense of the artificial bone [38]. As a final result, bioactive bone graft materials form a direct, living bond with the host bone. The healing periods used for our study, that is, 2, 4, and 8 weeks, were selected to demonstrate early and later stages of bone remodeling and to determine the progression of the bone while interacting with the different bone grafts.

An adverse correlation between the presence of TGF-β1 and bone formation was described in in vivo experiments [39,40]. Moreover, a fibrotic layer around the surface was found after the insertion of dental implants with added TGF-β1 [41]. In vitro experiments, however, proved that TGF-β1 suppressed the osteoblastic differentiation of bone marrow cells, whereas its inhibition released the stromal cells from their differentiation arrest, facilitating the formation of terminally differentiated osteoblasts [42]. Furthermore, TGF-β1 has been related to cellular processes for wound healing as well as for the regulation of inflammatory processes [21]. That fact led our group to explore the biofunctionalization of titanium dental implants with TGF-β1 inhibitor peptides, which resulted in accelerating the osteoprogenitor cell differentiation in vitro [30].

P144 (NH2-TSLDASIWAMMQN-OH) is a TGF-β inhibitor peptide developed by CIMA (University of Navarra) that excels in the arrest of the TGF-β1, inhibiting its interaction with fibroblast cells. Due to this inhibition, the osteoblast cells migrate to the calcium phosphate and consequently favor the osseointegration. P144 is a peptide derived from the TGF-β type III receptor. This peptide is in the cytoskeleton of many cell types. P144 interacts with TGF-β1 in the area where the cytokine connects with the cell receptor and blocks the cell signaling pathway [43,44,45].

Several techniques have been proposed for socket preservation after teeth extraction, although none has yet been proven as the gold standard. Moreover, a higher loss of bone can occur because of endodontic pathology, periapical granuloma infection, periodontitis, or traumatic extraction [10]. Bouwman et al. proved in 2010 the osteoconductive properties of BCP material histomorphometrically and histologically in a sinus floor experiment [4]. Bone maturation was evident by the presence of lamellar bone [10]. Our results prove that the biofunctionalization with P144 of a BCP biomaterial has an additive effect, since it stimulates both osteoconduction and osteoinduction by avoiding the loss of bone mass due to teeth extraction.

There is a notable histomorphometric similarity of beagle and human bone [9,46,47]. Despite the considerable variability in the methods to measure the contact between the osseous tissues and the profile of the implants [48,49,50], BIC (Bone index contact) represents a quantitative measurement that has been used by many authors as a measure for osseointegration. The observations with BS-SEM offer a simpler, less resource-consuming, and highly discriminative method for BIC determination [51], constituting a non-subjective, systematic measurement of the bone contact of all the active implant surface.

No standardized, objective, and systematic measurement has been proposed to date to assess the bone tissues in living contact with the biomaterial fragment’s surface. Alt et al. [52] and Ding et al. [53] propose that the histomorphometric analysis be made with histological images, but different parameters and regions of interest (ROIs) are proposed. Our analysis was made also on the whole perimeter of the alveolar defect as represented in a stitched complete image of the mandible.

The region of interest (ROI) was determined by using the profile of the alveolar cortical as a limit. For each sample, image segmentation and binarization with the ImageJ program was carried out. Then, the biomaterial, the osteogenesis, and the non-calcified tissues were quantified, as represented in Figure 13.

The ROI selected corresponded to the contour of the alveolar defect, excluding the bone tissues constituting the alveolar wall cortical profile. Then, the biomaterial granule’s area was measured and the new osseous tissues around the granules were quantified. The values that reached statistical significance were the new bone volume (NBV), defined as the percentage (%) of newly deposited bone tissue within the ROI, and NBV/Total volume (TVol), that is, the ratio of mineralized new bone to the ROI after subtracting the area corresponding to the biomaterial granules, estimated from the analyzed sections. The results of this quantitative analysis were obtained by stitching histological images, following a method similar to the one described by Bakhshalian et al. [54]. These authors excluded from their analysis the cortical bone, and subtracted the profile of the biomaterial’s granules for calculating the total volume of the ROI. Moreover, these calculations were made without taking into account valuable information that could eventually be applied to the analysis of the osseointegration of granulated biomaterials, such as the trabecular number (Tb.N), trabecular separation (Tb.Sp), trabecular thickness (Tb.Th) and especially the trabecular bone pattern factor (Tb.Pf) as reported by Blouin et al. [55] by using the “Standardized Nomenclature for Bone Histomorphometry” that was established by Parfitt in 1987 and Parfitt et al. [56,57] to evaluate bone remodeling [58].

Our histomorphometric results showed a better behavior of our P144-biofunctionalized grafts with respect to the untreated samples. However, our histological and BS-SEM observations demonstrated the importance of the contact between the preexisting alveolar cortical bone and the new bone tissues deposited around the biomaterial granules. The connections between the newly formed bone trabeculae are relevant to assess the quality of the bone filling the alveolar space and thus the bone anchorage for the future implant placement.

Our results at the early stages of healing (two weeks) showed higher values for the experimental group. While the control group showed a lineal increase of bone ingrowth, our experimental group showed decreased values at four weeks; this decrease could be explained by our qualitative results (BS-SEM and histology), where at higher magnification the peptide biofunctionalized grafts showed more vascular spaces within the alveolar defect. This neovascularization permits granules to become invaded by vessels and facilitates protein adsorption. Indeed, this can be observed by the color change of the fragments that adopted the same green hue as the osseous tissues around them.

Two weeks after implantation, the calcified tissues surrounding the biomaterials revealed by the light and electronic microscopy analysis could help to explain the numerical results of the analysis presented in Figure 1 and Figure 2. Thus, the osteoconductive properties of the biomaterial allowed the granules to approach the preexisting cortical; the granules were in contact with new bone starting to grow around them.

BS-SEM examination permits the observation of calcified tissues according to different levels of whiteness, which depends on their calcium concentration [50,59,60,61,62,63]. At four weeks, the biofunctionalized granules showed a dense bone structure in contact with the preexisting corticals; the granules were being integrated through the entire alveolar defect and the newly formed bone seemed more structured and compact than in previous weeks. These results are consistent with the numerical analysis presented in Figure 1 and Figure 2, taking into account that our initial quantitative analysis did not consider the preexisting cortical bone.

At eight weeks, the values of bone ingrowth were similar for both groups. These results confirm the promoting properties of the biofunctionalization technique, at least partially designed to attract mesenchymal stem cells in order to produce their phenotype to osteogenic cell lines. The backscattered scanning electron microscopy analysis of the alveolar defect sections depicted the bone ingrowth at the expense of the granules; trabeculae were thicker and lamellar bone apposition was denser in the biofunctionalized grafts.

The tissues of the first trabeculae that surround the granules as well as the ones formed as a response to the periosteal injury are mainly constituted by chondroid bone [64]. It has a faster calcification and contains types I and II collagen fibers [65,66]. Woven, lamellar, and osteonal bone are the calcified tissues that replace chondroid bone in our histological and BS-SEM findings, in a similar manner to the one seen in other endomembranous ossification-based processes, such as mandibular symphysis closure [62], skull suture growth and closure [67,68,69], tooth eruption [70,71], fracture repair [59,60,61,62,63,64,65,66,67], and, more recently, the osseointegration of non-metallic biomaterials [14,35].

Our histological analysis demonstrated the presence of lining cells in the surface perimeter of the granules at the early stages of osseointegration. The lining cells remained active in the samples at the end of the experimentation; the active deposition of non-calcified bone matrix in the border of the vascular channels and granules was observed in greater measure in the biofunctionalized bone grafts. The differences in the coloring of the lamellar bone showed the maintenance of the lamellar apposition throughout the experimental period.

The positive effects of biofunctionalization on the osseointegration of synthetic bone graft could be not only immediate but also in the medium or long term, thanks to trabecular apposition and lamellar remodeling, as described by Martino et al. [72] and Biguetti et al. [73]. Our TGF-β1 inhibitory peptides have proven to be able to adhere to Cp-Ti surfaces and keep their bioactivity [67] by evoking an in vitro cell response with a reduction of fibroblast differentiation and an increase of osteoblastic markers [30].

The heterogeneity of the morphology of the alveolar defects, surgical procedure, trauma, and duration have to be taken into account because they present an important influence on the resorption and biological response. The limited literature available with respect to TGF-β1 inhibitor peptides regarding bone organization obliges us to take these results with caution. Further research on biofunctionalization with bioactive peptides such as this TGF-β1 inhibitor peptide and its effects on osseointegration is necessary [36].

In relation to P144’s stability, Sevilla et al. [74] determined that once the calcium phosphates have the P144, the activity is immediate and the peptide is not denaturalized under normal conditions. The mechanical stability was determined by sonication. The samples were introduced in a solution of NaOH 0.1 M with streptavidin-FITC conjugate and sonicated until the surface released all the fluorescence. Fluorescence intensity of the samples was measured using an Olympus E600 upright microscope, equipped with a digital camera Olympus DP 25 (Olympus, Tokyo, Japan). The samples were sonicated for 30 h in water and the fluorescence intensity was measured after different sonication periods. The results showed a good mechanical stability of the biofunctionalized calcium phosphate, revealing that the peptides presented high bond forces due to the covalent character of the bond to the calcium phosphate after 30 h of sonication [70].

## 4. Materials and Methods

### 4.1. Description of Bone Grafts

A synthetic bone graft was used for this study and biofunctionalization was carried out for comparison, namely, (i) Maxresorb^®^ Botiss Klockner (*n* = 12) and (ii) Maxresorb^®^ Botiss Klockner biofunctionalized with P144 peptide (*n* = 12). A schematic comparison can be seen in Figure 14. Maxresorb^®^ is a 100% synthetic bone graft substitute with controlled resorption properties. This biomaterial shows a homogenous and biphasic composition of 60% hydroxyapatite (HA) and 40% beta-tricalcium phosphate (β-TCP) [60,61,62,63].

Dilution of P144 peptide in sterile PBS (1 mmol/mL concentration) was done. Afterward, a 0.5 mL portion was added to 2 mg of Maxresorb^®^ and mixed just before introducing the graft into the alveolar defects. The concentration of the peptide inside the biomaterial was 10 mg/cc.

### 4.2. Animal Model and Animal Selection Criteria

For this research, six one-year-old male beagle dogs (body weight 12–15 kg) were used. The project was approved by the University Ethical Committee (CEEHA 3256 DMAH 8890) and all animal procedures were performed under the European Community Guidelines (Directive 2010/63/EU). Prior to surgery, animals were acclimatized to the local environment for two weeks.

### 4.3. Surgical Procedures

All experimental surgeries were performed at the Hospital Universitari de la Facultad de Veterinaria of the Universitat Autònoma de Barcelona (Barcelona, Spain).

Regarding the surgical process, beagle dogs were preanesthetized using medetomidine and methadone. The anesthesia used was propofol and diazepam injected intravenously. The dogs were maintained with inhaled isoflorane in an oxygen carrier. During the surgical operation, all dogs received an intravenous saline isotonic solution and an intravenous cefazolin injection. Local dental anesthesia was achieved with articaine 4% 1:100,000 epinephrine using the infiltrative technique at the site of extractions and surgery in the mandibles.

Root canal treatment of second and third premolars (P2, P3) was carried out two months prior to surgery. Extractions of both premolars of each hemimandible were carried out following an atraumatic technique and bone grafts were inserted in the alveolar defects at the same surgery.

Flap margins were adapted using a tension-free suture with bioresorbable polyglactin 910 Vycril©4/0.

After surgery, all dogs received cefovecin subcutaneously as a prophylactic long-acting antibiotic as well as methadone and meloxicam as pain relief. A non-steroidal anti-inflammatory drug was given subcutaneously to the animals for a week to prevent inflammation and pain during the postoperative time. The healing was uneventful and peri-implant mucosa appeared to be clinically healthy [75].

Two dogs were euthanized at 2, 4, and 8 weeks after implantation, by an intravenous injection of an overdose of pentobarbital sodium. Before the euthanasia, the dogs were sedated by medetomidine in order to see to the animals’ welfare.

### 4.4. Histology and Sample Preparation

All samples were processed following the same protocol [32]. To preserve tissues, mandibles were immediately immersed in a 10% formalin solution (pH = 7) for three weeks. Afterward, the jaws were dehydrated by immersion in aqueous solutions with increasing ethanol concentrations (30%, 50%, 70% 96%, and twice 100%). Once totally dehydrated, samples were embedded in MMA resin Technovit 7200 VLC (Kulzer-Heraeus, Hanaus, Germany) by immersion with increasing resin concentrations (30%, 50%, 75%, and 100%), using benzoic peroxide as initiator.

Then, resin polymerization was carried out with a photopolymerization unit EXAKT 520 (Exakt), for 12 h with white light and 12 h with UV light. After the polymerization process, samples were stored in a heater (37 °C) to evaporate resin vapors and cut in coronal sections using an irrigated diamond saw EXAKT 300 (Exakt, Oklahoma City, OK, USA). All samples were polished by a grinding machine EXAKT 400 CS (Exakt). Then, the block samples were sectioned into two blocks. One block was sectioned and the slices polished using SiC abrasive papers with different index mesh (P800, P1200, and P4000, Buelher, Lake Bluff, IL, USA) until a thickness of 50 µm was obtained, and then stained with Masson-Goldner trichrome (Merk, Barcelona, Spain). Slices from all samples were processed for optical microscopy; the alternate samples from the blocks selected for further analysis were reprocessed to be observed by backscattered scanning electron microscopy (BS-SEM). The half-blocks were polished with the same SiC papers and coated with carbon sputtering to be observed under scanning electron microscopy (SEM, Jeol J-6510^®^, Tokyo, Japan).

### 4.5. Image Acquisition and Quantification

Optical microscopy was performed using a Nikon E600 optical microscope (Nikon, Tokyo, Japan) at different magnifications (×5, ×10, ×20, ×50, ×100). A stitching procedure was made to obtain an image of the whole sample for quantification. Samples were observed at higher magnifications for quality and cell identification.

Quantification was carried out with Photoshop (2017, Adobe Systems, San José, CA, USA) and ImageJ (150.i, National Institutes of Health (NIH), Bethesda, MD, USA) to determine the quantity of osteogenesis at the zone of the alveolar defect. The region of interest (ROI) was determined by using the profile of the alveolar cortical as a limit. For each sample, by image segmentation and binarization, the biomaterial, the new bone, and the non-calcified tissues were quantified. NBV (%) was defined as the area occupied by new bone within the ROI; TVol (%) was the result of the subtraction of the area corresponding to the biomaterial from the ROI. NBV/TVol (%) was the ratio of mineralized new bone to the total alveolar volume estimated from the analyzed section.

### 4.6. Statistical Analysis

The statistical study was carried out using the linear mixed model, in which the repeated factor was the time since surgery (2, 4, and 8 weeks) and the between factor was the biomaterial, the hemimandible being the covariate of no interest. We used a scaled identity covariance matrix. Each dependent variable was analyzed separately. For the percentage variables (proportions), we used the logit transformation. The Bonferroni correction was applied in order to control the effect of multiple comparisons, when it was necessary. Means and standard error of the mean were provided after adjusting for the covariate of no interest. All the analysis was carried out using SPSS V20.0 (IBM, New York, NY, USA). 

## 5. Conclusions

The parameters, values, and calculations proposed to assess bone biomaterial granules’ osseointegration must be reviewed. In terms of blood vessels (vascular channels) and bone tissues present in the early stages of healing (2 and 4 weeks) and remaining active in direct contact with the grafts until the end of the experiment, the biofunctionalized samples presented better results than the control ones. The biomaterial biofunctionalization maintained a stable endomembranous bone formation, more advanced than in the control samples, throughout the experiment timeline. The peptide biofunctionalization may shorten the healing period of osseointegrated biomaterials. The results of this study may serve as a useful reference for further studies oriented toward the analysis of the mechanisms behind biofunctionalization.

## Figures and Tables

**Figure 1 materials-12-03168-f001:**
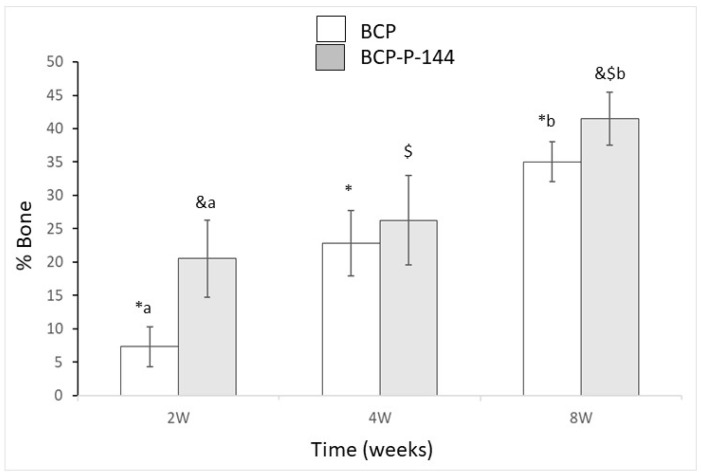
New bone volume (NBV) of newly formed bone within the region of interest (ROI) at 2, 4, and 8 weeks after bone grafting. BCP, biphasic calcium phosphate; BCP-P-144, biphasic calcium phosphate treated with P144. Statistically significant differences are indicated as *, #, and & for the different implantation times and with a or b for the different biomaterials BCP or BCP-P-144.

**Figure 2 materials-12-03168-f002:**
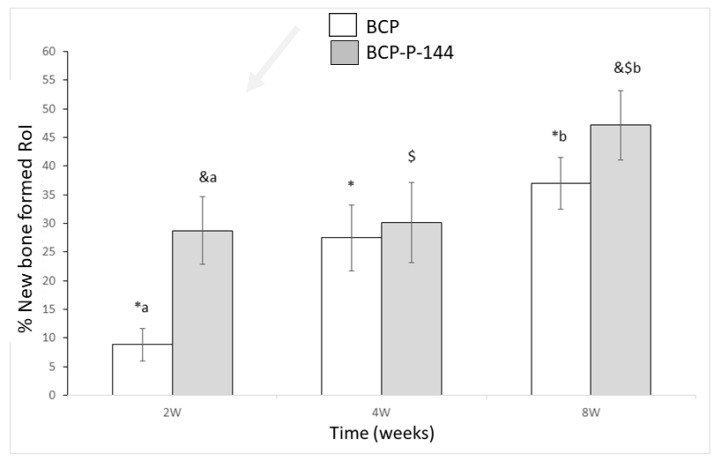
NBV/Total volume (TVol) of newly formed bone within the region of interest (excluding the area occupied by the biomaterial) at 2, 4, and 8 weeks after bone grafting. BCP, biphasic calcium phosphate; BCP-P-144, biphasic calcium phosphate treated with P144. Statistically significant differences are indicated as *, #, and &.

**Figure 3 materials-12-03168-f003:**
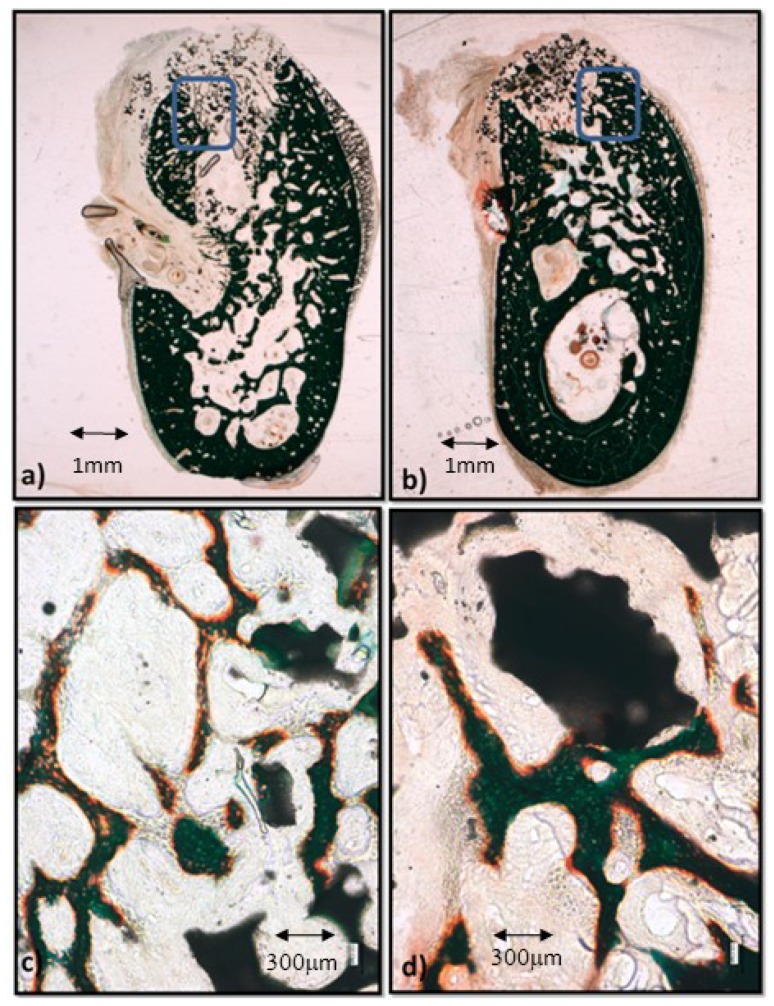
Histologies after two weeks of bone grafting insertion. (**a**) Control group sample; (**b**) P144-biofunctionalized bone graft; (**c**) control group showing the framed area in (**a**); (**d**) P144-biofunctionalized bone graft showing the framed area in (**b**).

**Figure 4 materials-12-03168-f004:**
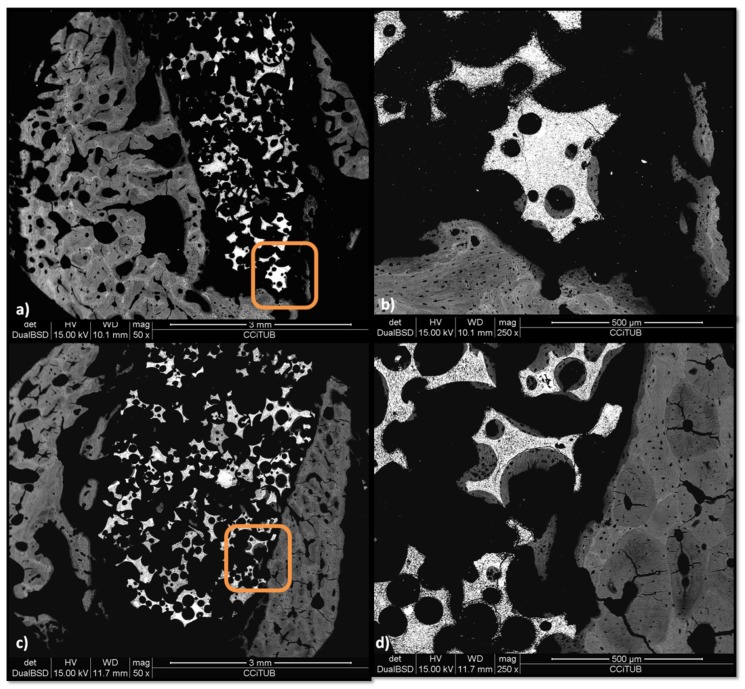
Backscattered scanning electron microscopy (BS-SEM) results after two weeks of bone grafting insertion. (**a**) Control group sample; (**b**) framed area in (**a**), newly formed bone around one granule; (**c**) P144-biofunctionalized bone graft; (**d**) framed area in (**c**), newly formed bone in contact with the biofunctionalized granules near the preexisting cortical.

**Figure 5 materials-12-03168-f005:**
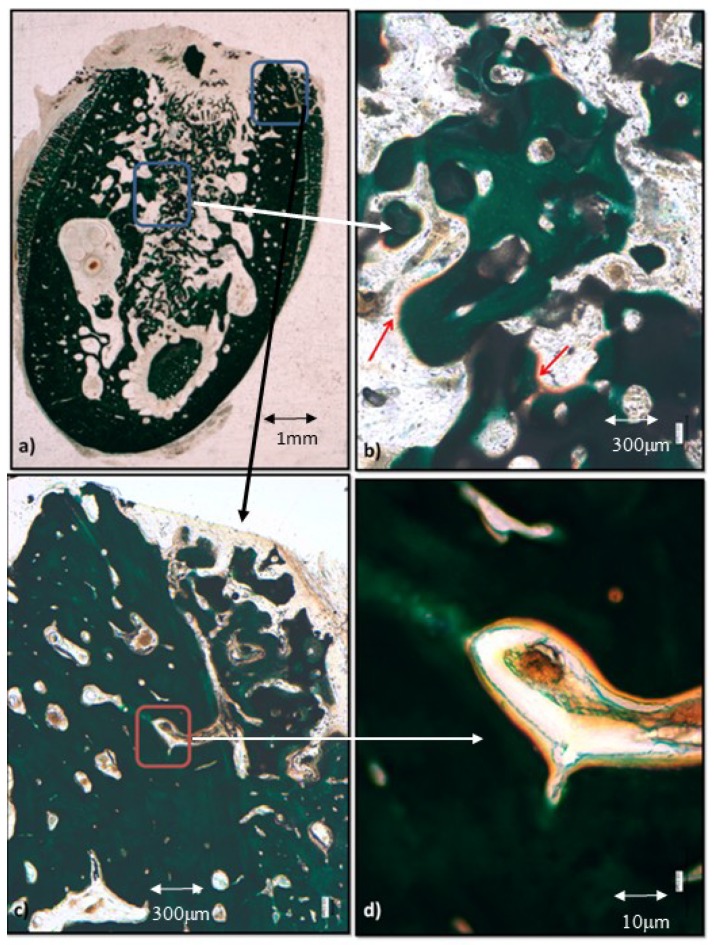
Results after four weeks of bone grafting insertion in the control group. (**a**) Stitching summative images of the control sample; (**b**) framed area in (**a**), orange (red arrows) non-calcified extracellular matrix (ECM) near the granules; (**c**) framed area in (**a**): vascular channel crossing the preexisting cortical; (**d**) framed area in (**c**), vascular channel.

**Figure 6 materials-12-03168-f006:**
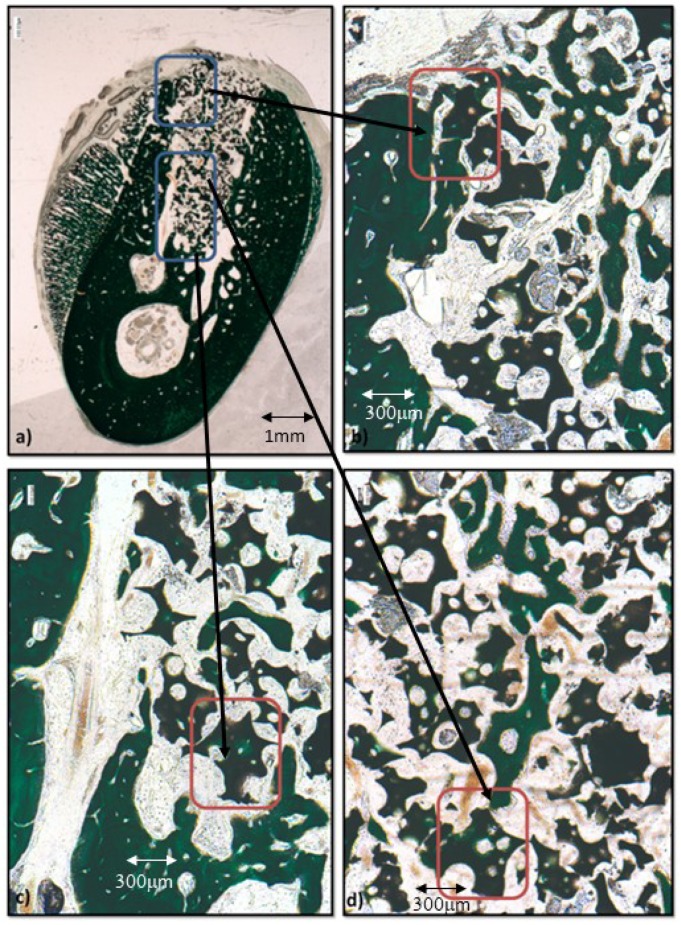
Results after four weeks of bone grafting insertion in the experimental group. (**a**) Stitching summative images of the P144-biofunctionalized sample; (**b**) framed area in (**a**), granules in contact with the cortical of the alveolar defect; (**c*,*d**) framed area in (**a**), newly formed bone around the granule (see series 1 and 2).

**Figure 7 materials-12-03168-f007:**
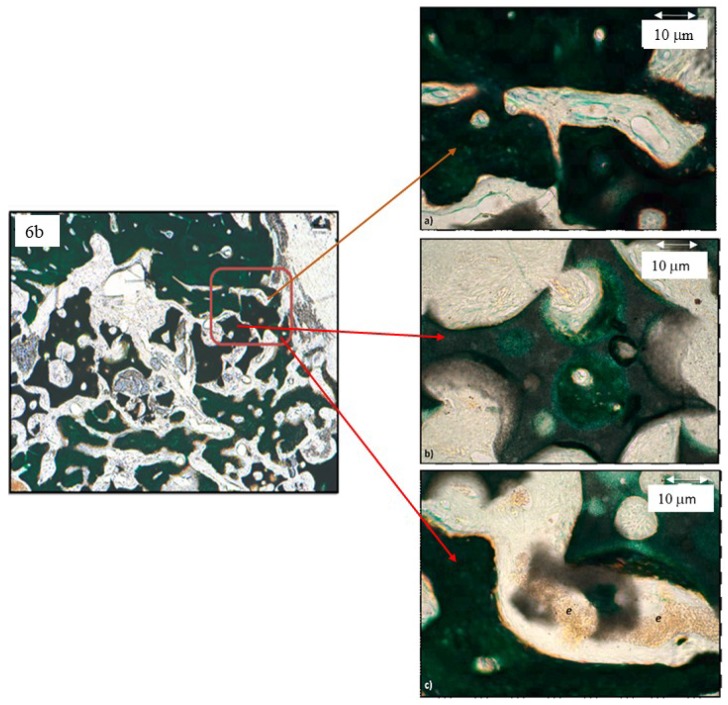
Series 1. Results after four weeks of grafting bone insertion, framed areas in Figure 6b. (**a**) Higher magnification area of Figure 6b indicated by the arrow. (**b**) Higher magnification area of Figure 6b indicated by the arrow. (**c**) Higher magnification area of Figure 6b indicated by the arrow with abundant vessels with erythrocytes (e).

**Figure 8 materials-12-03168-f008:**
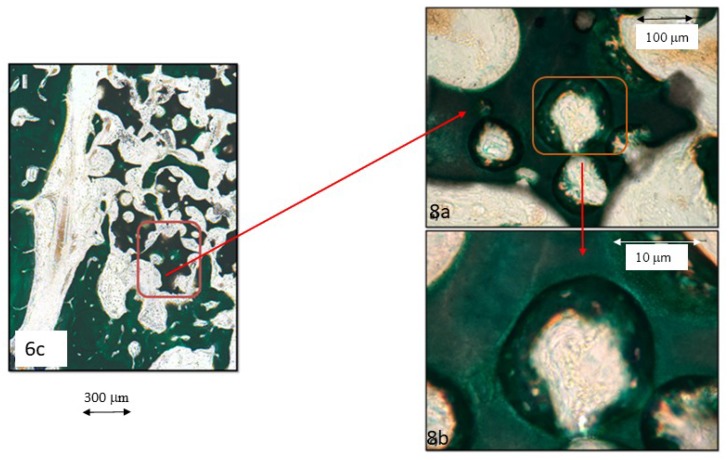
Series 2. Results after four weeks of grafting bone insertion, framed areas in Figure 6c of a P144-biofunctionalized sample. (**a**) Framed area in Figure 6c; (**b**) framed area in Figure 8a: newly formed bone growing from the granule.

**Figure 9 materials-12-03168-f009:**
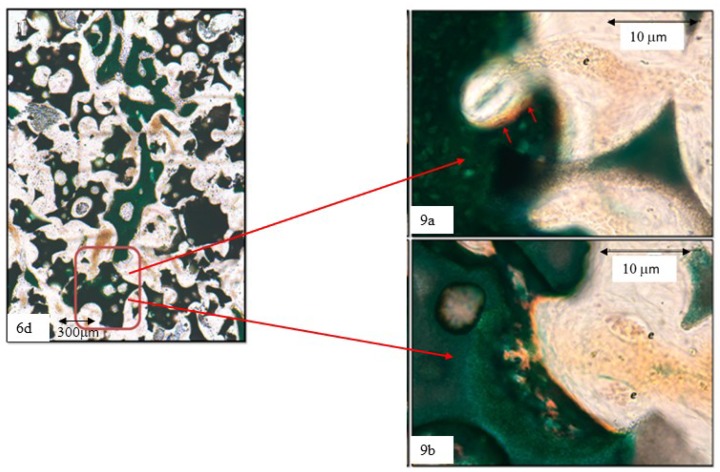
Series 2. Results after four weeks of grafting bone insertion, framed areas in Figure 6d of a P144-biofunctionalized sample. (**a**) Detail of the framed area in Figure 6d, osteoblasts (red arrows) inside the non-calcified EMC, erythrocytes (e). (**b**) Detail of the framed area, cell lacunae with type II non-calcified collagen and matrix deposition around the granule, abundant vessels with erythrocytes (e).

**Figure 10 materials-12-03168-f010:**
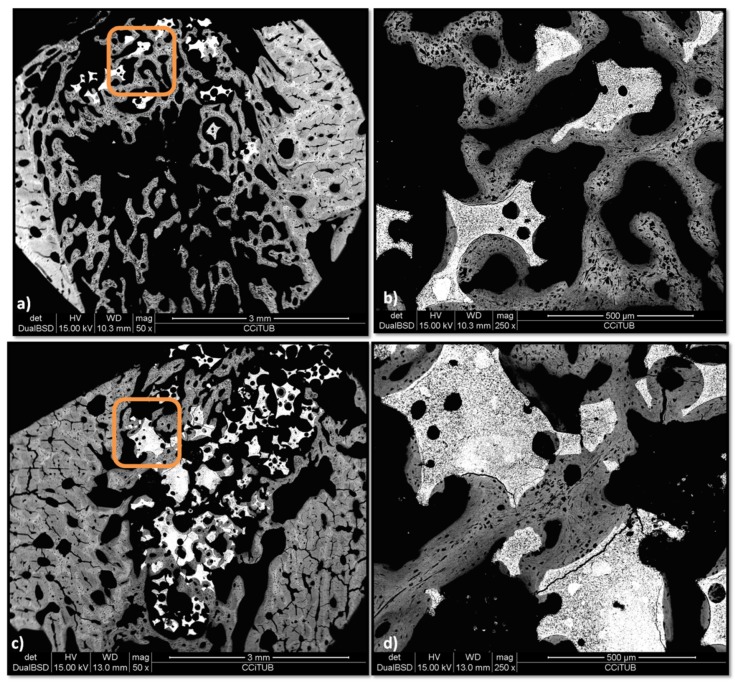
BS-SEM results after four weeks of bone grafting insertion. (**a**) Control group sample. (**b**) Framed area in Figure 9a, connected trabeculae with newly formed lamellar bone around a chondroid bone core. (**c**) P144-biofunctionalized bone graft. (**d**) Framed area in Figure 9c, interconnected mature lamellae of newly formed bone in contact with the biofunctionalized granules near the preexisting cortical.

**Figure 11 materials-12-03168-f011:**
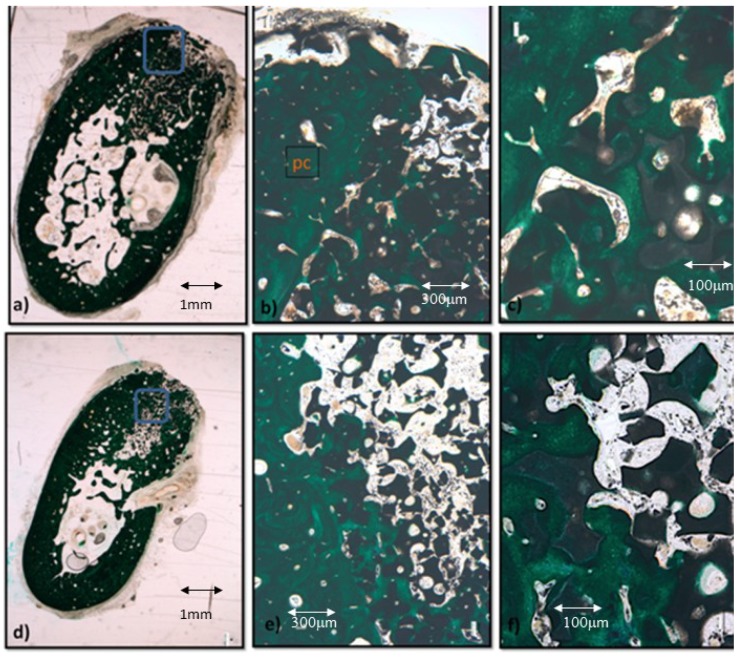
Results after eight weeks of bone grafting insertion. (**a**) Control group sample. (**b**) Framed area in Figure 10a; pc, preexisting cortical. (**c**) Higher magnification of Figure 10b. (**d**) P144-biofuncionalitzed sample. (**e**) Framed area in Figure 10d. (**f**) Higher magnification of Figure 10e.

**Figure 12 materials-12-03168-f012:**
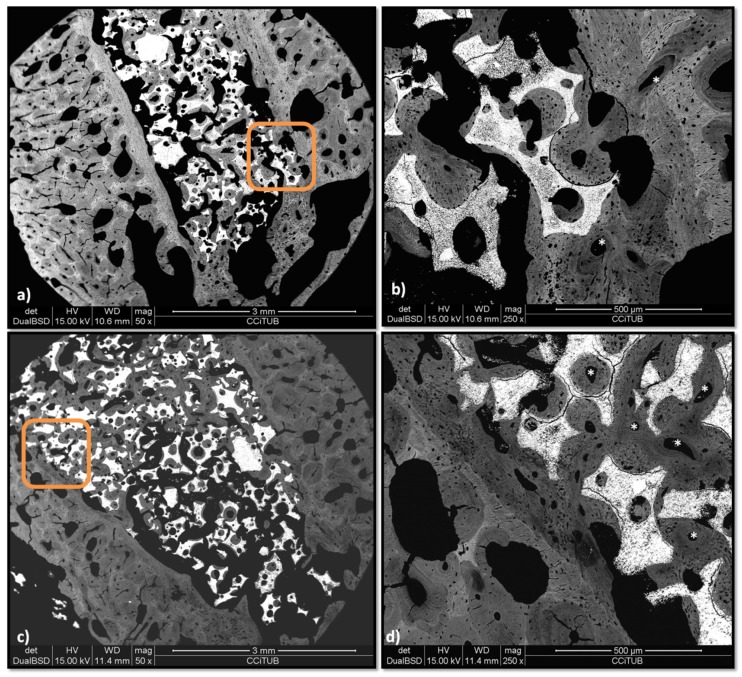
BS-SEM results after eight weeks of bone grafting insertion. (**a**) Control group sample. (**b**) Framed area in Figure 11a, * osteonal bone tissue. (**c**) P144-biofunctionalized bone graft. (**d**) Framed area in Figure 11c, * osteonal bone tissue.

**Figure 13 materials-12-03168-f013:**
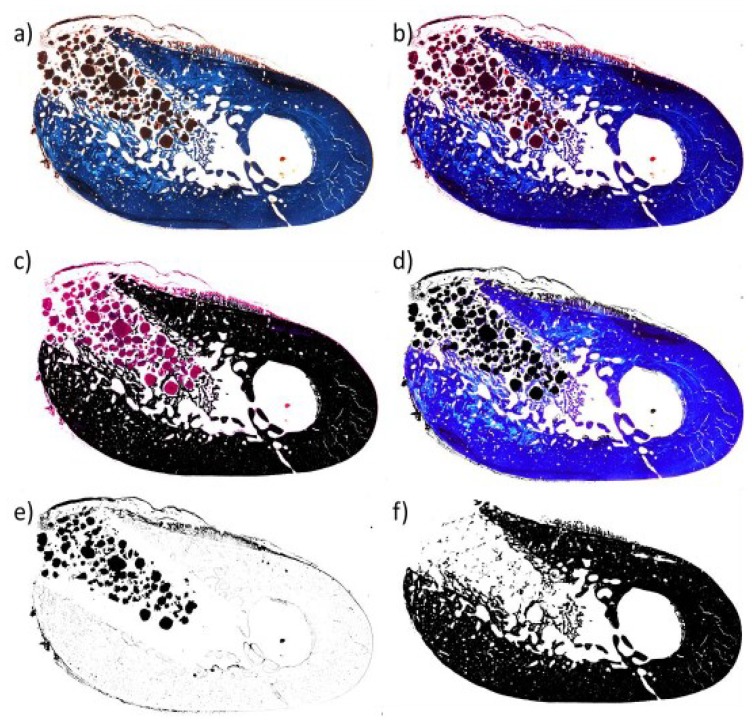
Image processing of biomaterials for quantification using ImageJ and determination of ROI. (**a**) Original image; (**b**) contrasted image; (**c**) bone tissue segmentation; (**d**) segmentation of the grafted biomaterial; (**e**) image binarization of (**d**); (**f**) image binarization of (**c**).

**Figure 14 materials-12-03168-f014:**
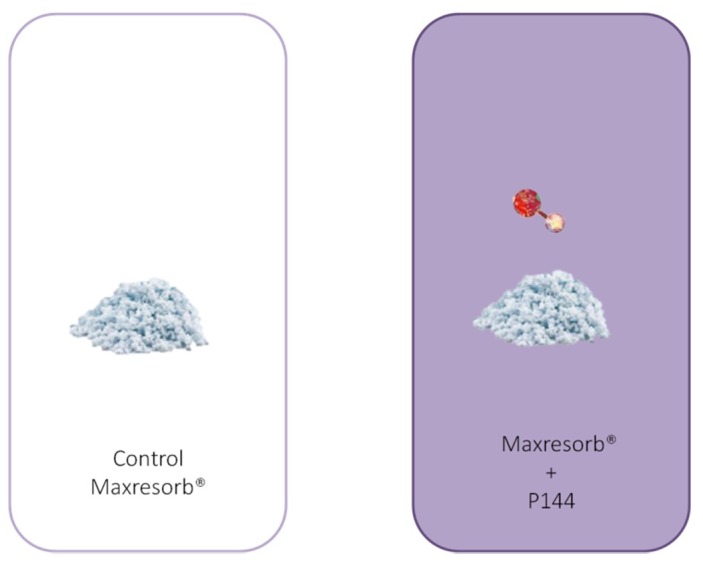
Comparison between calcium phosphate Maxresorb^®^ and the same calcium phosphate biofunctionalized with P144 in relation to osseointegration.

**Table 1 materials-12-03168-t001:** Bone index contact for biphasic calcium phosphate (BCP) and for bone calcium phosphate biofunctionalized with P-144 (BCP-P-144). Standard deviation in parentheses.

Biomaterial	2 weeks	4 weeks	8 weeks
BCP	20% (8)	33% (8)	65% (9)
BCP-P-144	30% (6)	40% (7)	77% (8)

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
