# Peer review of "Biofunctionalization with a TGFβ-1 Inhibitor Peptide in the Osseointegration of Synthetic Bone Grafts: An In Vivo Study in Beagle Dogs"

_materials, 2019, doi:10.3390/ma12193168_

Round 1

Reviewer 1 Report

The aim of this report is to clarify the effects of P144-biofunctionalized synthetic bone on tooth extraction socket healing.

I have some questions regarding the experimental methods and interpretation of the data.

Please see my comments below:

Overall

1.      What does it mean to be the term "osseointegration" in this manuscript?

Generally, osseointegration indicates the functional connection between bone and the surface of implant. Please explain the point.

2.      What are the mechanisms of functionalizing the Maxresorb with P144?

3.      How did the authors determine the concentration of P144?

4.      How long the P144 could be active after grafting?

Results section

1.      The structure of graphs is unfriendly for readers. In Figure 1 and 2, what do “K” and “KP” mean? Not only these points, please reconsider the overall graph structure.

2.      The authors should explain the Region of Interest (ROI) in detail, adding the new one figure for ROI.

Materials and methods section

1.      Please add P144 information including manufacturer and the purity of peptides.

Minor comments:

·        Please insert scale bars into the representative images in each figure.

·        Please add the schematic images of the mechanisms of Maxresorb biofunctionalization with P144.

Author Response

Dear Reviewer,

Thanks for taking the time to review our manuscript and suggest to us to improve our work by providing a lot more detail. We have done so, and we are now submitting a manuscript that not only addresses the points the you specifically raised but also many others that we have considered in order to deliver what we think is a much improved version of our work. This version includes a lot more paragraphs in all main sections, restructured subsections, new figures, and a new discussion and details to better reflect the contents of our contribution. We are looking forward to your comments.

Sincerely,

Francisco-Javier Gil Mur

You will enclosed the word document with the comments.

Thank you very much for your attention

Reviewer 2 Report

This manuscript, entitled ‘Biofunctionalization with a TGFβ-1 inhibitor peptide in the osteointegration of synthetic bone grafts: an in vivo study in beagle dogs,’ is considered to be interesting. However, this manuscript has many issues to be addressed. Furthermore, this manuscript is not considered to be prepared enough to be submitted to this journal.

The authors use the terms, ‘osteointegration,’ and ‘osseointegration’ simultaneously. The reviewer recommends that the authors should use one term, ‘osseointegration’ consistently.

Line 79.

   What does it mean by ‘TGB superfamily members’? There is no full name about ‘TGB’ in this manuscript.

Figures 1 and 2.

   The reviewer would like to say that this figure is insincere to the readers of this journal. What do ‘K’ and ‘KP’ stand for? What are the vertical lines? If the vertical lines are the days of sacrifice, why do the authors express the intervals between weeks 2 and 4, and between 4 and 8 as similar spaces?

Figure 3.

The framed areas do not match the magnified images. Also, the reviewer would like to say that the figure legend is generally described as sentences, not phrases. Please, reorganize the description in the body of text and the figure legend. Some descriptions are more adequate for the figure legend, not for the body of text. Additionally, the subtitles should follow the journal format.

Figure 4.

   Please, be consistent in organizing the figures. In Figure 3, (a) and (b) are less magnified than (c) and (d). On the other hand, in Figure 4, (a) and (c) are less magnified than (b) and (d). Also, please, reorganize the figures more clearly using some indicators like arrows.

Figure 5.

   Which framed area is shown in Figure 5b? There are two framed areas in Figure 5a.

Figure 6-11.

   Please, reorganize the figures. The framed areas do not match at all. Even in some figures, the figure legends do not match the figures. Actually, the aspect ratio in a framed area should be equal to that in its magnified image. The authors are considered to prepare the figures following such a rule. Therefore, every figure of this manuscript is confusing and ambiguous.

Actually, the reviewer regrets to say that there are so many other issues to be addressed than the figures. Therefore, this manuscript is considered to be definitely rejected. The reviewer recommends that the authors should be thoughtful in preparing the manuscript for clear description and organization.

Author Response

Dear Reviewer,

Thanks for taking the time to review our manuscript and suggest to us to improve our work by providing a lot more detail. We have done so, and we are now submitting a manuscript that not only addresses the points the you specifically raised but also many others that we have considered in order to deliver what we think is a much improved version of our work. This version includes a lot more paragraphs in all main sections, restructured subsections, new figures, and a new discussion and details to better reflect the contents of our contribution.

We are looking forward to your comments.

Sincerely,

Francisco-Javier Gil Mur

PD. You will find enclosed a word document with the authors' answers.

Thank you very much for your attention

Reviewer 3 Report

Maxresorb bone graft should be described in greater detail . Dfdba (?) or Fdb a (?) makes a difference in regeneration. Also variation in resorption properties 

Author Response

Dear Reviewer,

Thanks for taking the time to review our manuscript and suggest to us to improve our work by providing a lot more detail. We have done so, and we are now submitting a manuscript that not only addresses the points the you specifically raised but also many others that we have considered in order to deliver what we think is a much improved version of our work. This version includes a lot more paragraphs in all main sections, restructured subsections, new figures, and a new discussion and details to better reflect the contents of our contribution.

Following your comments, the authors have incorporated new aspects and characteristics about the calcium phosphate Maxresorb in order to explain the osseointegration behavior.

We are looking forward to your comments.
Sincerely,

Francisco-Javier Gil Mur

Round 2

Reviewer 2 Report

Thank you for accepting the review comments and for making an effort in revising the manuscript. This manuscript is considered to be much improved. However, there are still many concerns in this manuscript. For example, there is no framed area in Figure 4b or Figure 4d. Therefore, Figure 7 is meaningless. Also, the reviewer does not understand Figure 8 at all because which area in Figure 4 is magnified cannot be seen.

Conclusively, the reviewer regrets to say that this manuscript is not acceptable for publication of this journal. The reviewer hopes that the authors understand that this decision is very hard to be made.

Author Response

Dear Reviewer:

First of all, I want to thank very sincerely the comments of the reviewers who have contributed to improve the quality of the contribution. The reviewer is right. The authors have made an important mistake in the figure legends that avoided understanding the text. We apologize for nor correcting in the second version.

The authors have answered and clarified all the points that have been discussed and we hope that in this new version the paper can be published in the Materials.

Thank you veru much

FJ Gil

Full Professor
